# MicroRNAs Signature Panel Identifies Heavy Drinkers with Alcohol-Associated Cirrhosis from Heavy Drinkers without Liver Injury

**DOI:** 10.3390/biology12101314

**Published:** 2023-10-08

**Authors:** Fathima Shihana, Mugdha V. Joglekar, Tae-Hwi Schwantes-An, Anandwardhan A. Hardikar, Devanshi Seth

**Affiliations:** 1The Centenary Institute of Cancer Medicine & Cell Biology, The University of Sydney, Sydney, NSW 2006, Australia; 2Edith Collins Centre (Translational Research in Alcohol Drugs and Toxicology), Sydney Local Health District, Sydney, NSW 2050, Australia; 3Diabetes & Islet Biology Group, School of Medicine, Western Sydney University, Campbelltown, NSW 2560, Australia; m.joglekar@westernsydney.edu.au (M.V.J.); a.hardikar@westernsydney.edu.au (A.A.H.); 4Department of Medical and Molecular Genetics, Indiana University, Indianapolis, IN 46202, USA; tlschwan@iu.edu; 5Sydney School of Medicine, Faculty of Medicine and Health, The University of Sydney, Sydney, NSW 2006, Australia

**Keywords:** alcohol-associated liver cirrhosis, microRNAs, biomarker

## Abstract

**Simple Summary:**

Alcohol-associated liver disease (ALD) is a preventable disease if identified early, but current biomarkers lack specificity to identify who amongst drinkers will progress to advanced disease, i.e., cirrhosis. MicroRNAs are small molecules that play a regulatory role in several diseases, are affected by alcohol and may be important players in alcohol use disorders, such as ALD. So far, microRNAs associated with ALD are based on comparisons with non-drinking healthy controls. Our new approach was to compare global miRNA profiles in heavy drinking patients without liver injury versus heavy drinkers with cirrhosis to identify miRNAs specific to liver disease. Our novel discoveries include (i) the identification of a set of microRNAs that differentiated drinkers with cirrhosis from those without liver injury; (ii) an inverse relationship of microRNAs expression with disease severity, suggesting their potential use in monitoring disease progression; (iii) the identification of microRNA target pathways involved in cell death, inflammation, fibrosis and fat metabolism; (iv) and the association of some microRNAs with known genetic risks involved in triglyceride metabolism, underscoring the growing role of lipids (fats) in ALD. This study advances knowledge in the field by identifying new microRNAs as potential diagnostic and/or therapeutic targets.

**Abstract:**

**Background:** Alcohol-associated liver disease (ALD) is the most common disorder of prolonged drinking. Mechanisms underlying cirrhosis in such patients remain unclear. MicroRNAs play regulatory role in several diseases, are affected by alcohol and may be important players in alcohol use disorders, such as cirrhosis. **Methods:** We investigated serum samples from heavy chronic alcohol users (80 g/day (male) and 50 g/day (female) for ≥10 years) that were available from our previously reported GenomALC study. A subset of GenomALC drinkers with liver cirrhosis (cases, n = 24) and those without significant liver disease (drinking controls, n = 23) were included. Global microRNA profiling was performed using high-throughput real-time quantitative PCR to identify the microRNA signatures associated with cirrhosis. Ingenuity Pathway Analysis (IPA) software was utilized to identify target mRNAs of significantly altered microRNAs, and molecular pathways were analysed. Identified microRNAs were analysed for correlation with traditional liver disease biomarkers and risk gene variants previously reported from GenomALC genome-wide association study. **Results**: The expression of 21 microRNAs was significantly downregulated in cases compared to drinking controls (*p* < 0.05, ∆∆Ct > 1.5-fold). Seven microRNAs (miR-16, miR-19a, miR-27a, miR-29b, miR-101, miR-130a, and miR-191) had a highly significant correlation (*p* < 0.001) with INR, bilirubin and MELD score. Three microRNAs (miR-27a, miR-130a and miR-191) significantly predicted cases with AUC-ROC 0.8, 0.78 and 0.85, respectively (*p* < 0.020); however, INR performed best (0.97, *p* < 0.001). A different set of six microRNAs (miR-19a, miR-26a, miR-101, miR-151-3p, miR-221, and miR-301) showed positive correlation (ranging from 0.32 to 0.51, *p* < 0.05) with rs10433937:*HSD17B13* gene variant, associated with the risk of cirrhosis. IPA analysis revealed mRNA targets of the significantly altered microRNAs associated with cell death/necrosis, fibrosis and increased steatosis, particularly triglyceride metabolism. **Conclusions**: MicroRNA signatures in drinkers distinguished those with liver cirrhosis from drinkers without liver disease. We identified mRNA targets in liver functions that were enriched for disease pathogenesis pathways.

## 1. Introduction

Alcohol-associated liver disease (ALD) is one of the most common liver diseases in the world and the major medical consequence of excessive drinking. Excessive alcohol consumption can lead to the development of fatty liver, alcoholic hepatitis and fibrosis that progresses to cirrhosis in up to 20% of patients [1]. Furthermore, if the damage is not controlled, more than 1.5% of people with cirrhosis etiology can progress to hepatocellular carcinoma (HCC) and high mortality [2].

The clinical manifestations of the patients with cirrhosis due to ALD widely range from asymptomatic to various decompensation events [3]. However, the definite diagnosis of ALD is based on a liver biopsy, which has drawbacks such as being invasive, having limited tissue scope, and being inappropriate for monitoring progression and the assessment of steatosis being somewhat subjective [4]. Early diagnosis should help with the prevention of disease progression and treatment. Single-nucleotide polymorphisms (SNPs) have been identified as genetic risk factors for alcohol-associated cirrhosis in recent genome-wide association studies (GWAS) [5,6]. These risk SNPs are useful to predict ‘inherent’ risk; however, expression-based biomarkers are useful for understanding disease progression in real time. The need for novel non-invasive diagnostic biomarkers, including microRNAs [3], in ALD is necessary.

Apart from diagnostic capabilities, microRNAs also regulate the gene expression of SNPs and at the post-transcriptional level by promoting mRNA degradation or inhibiting mRNA translation and exert important biological functions in various pathological and cellular pathways [7,8,9]. Circulating microRNAs have been proposed as biomarkers in cholangiocarcinoma, hepatocellular carcinoma, non-alcoholic steatohepatitis (NASH) and cirrhosis [10,11,12]. Therefore, identifying circulating microRNAs could be useful biomarkers to both stratify patients with advanced chronic liver disease and early detection of ALD. This study profiled global circulating microRNAs in heavy alcohol users with an aim to identify microRNA biomarker signatures associated with ALD and study their role in ALD pathogenesis.

## 2. Methods

### 2.1. Human Samples

Frozen serum samples (−80 °C) were available through our multinational GenomALC study for an Australian sub-cohort from multiple centres in Sydney recruited between 2011 and 2017. Recruitment and cohort characteristics have been detailed previously [13,14]. Briefly, participants with long-term heavy alcohol use (80 g/day (male) and 50 g/day (female) for ≥10 years (n = 47) were included in the study. Cases (n = 24) are drinkers with unequivocal evidence (clinical/histological/imaging) of cirrhosis. Drinking controls (n = 23) are drinkers with no clinical/histological/imaging evidence of significant liver disease (i.e., no fibrosis) detailed in our previous GenomALC reports [13,14]. Cohort characteristics are described in Table 1. The Model for End-Stage Liver Disease (MELD) was calculated using The MELD Calculator available on US Department of Health and Human Services website (https://optn.transplant.hrsa.gov/data/allocation-calculators/meld-calculator/, accessed on 21 February 2010). The study was approved by the Human Research Ethics Committee (X11-0072, HREC11RPAH88).

### 2.2. RNA Extraction

Total RNA (including small RNA species) was isolated from 100 µL serum using the RNeasy-HT Kit (Qiagen, Hilden, Germany) according to the manufacturer’s instructions and QiaCube-HT robotic RNA isolation system. Each serum sample was mixed with 500 µL TRIzol (ThermoFisher Scientific, Waltham, MA, USA), and then spiked with 2.5 µL of 50 nM synthetic control microRNA ath-miR-172a (Sigma Aldrich, Hamburg, Germany). All procedures were performed according to the manufacturer protocol instructions. The quantity and quality of total RNA was measured using the NanoDrop Spectrophotometer, and RNAs with a 260/280 ratio greater than 1.6 was considered acceptable.

### 2.3. OpenArray^®^ Panels—Quantitative Real-Time PCR

cDNA was synthesised from total RNA and preamplified for qPCR OpenArray (TaqMan^®^ OpenArray^®^, ThermoFisher Scientific, USA) using protocols described in detail previously [15,16].

### 2.4. Data Analysis

High-throughput data generated from TaqMan^®^ OpenArray^®^ qPCR data were uploaded into ThermoFisher iCloud for the global normalisation and detection and determining the cut-off threshold. microRNAs that were undetectable or with an Amp Score < 1.24 and quantitation cycle (Cq) confidence < 0.6 were eliminated from further analysis as per the pre-defined quality control criteria for this study [15,16]. The detection threshold cycle (Ct) cut-off was defined at Ct value of 39.

### 2.5. Statistical Analysis

The relative fold change (ΔΔCt = ΔCt drinking control − ΔCt patient sample) between each group. Statistical significance (*p*-values) calculated using Student’s *t*-test was used to compare two groups, while for more than two groups analysis of variance (ANOVA) test was used in Excel. Regression/correlation analysis (Prism and Excel) was performed using significantly different microRNAs with traditional biomarkers and allele dosage of risk variant information from our genotype data [5]. The statistical program R was used to plot heat maps to visualise differentially expressed microRNAs. Volcano plots were plotted using the GraphPad Prism 9 software (GraphPad, San Diego, CA, USA). The Mann–Whitney test was used to compare the differences in microRNA expression between patients and drinking controls. *p*-values below 0.05 were considered statistically significant for all tests.

### 2.6. Pathway Analysis of the Dysregulated microRNAs

We used Ingenuity Pathway Analysis (IPA) software version 01-22-01 (www.qiagenbioinformatics.com) and microRNA target filter to predict target mRNAs of the differentially expressed microRNAs as described [17]. For identifying disease and function pathways, target mRNAs were overlaid on ‘Toxic’ and ‘Functions’ derived from select microRNAs using ‘Pathway design’ option in IPA. The top ten toxic (Tx) and functional (Fx) pathway networks, identified for microRNA-mRNA interactions, were plotted. We also used co-analysis function to identify significantly enriched pathways that are associated with the microRNAs.

## 3. Results

### 3.1. Cohort Characteristics

Of the 47 heavy alcohol users (n = 23 cases, n = 24 drinking controls), more than 99% were European Caucasian. Drinking controls and cases had similar median age (51 vs. 53) and proportion of males (66% vs. 65%), respectively (Table 1). As expected, cases with liver cirrhosis had significantly different liver enzymes and liver functions compared to the drinking controls: lower haemoglobin level, platelet count and serum albumin, and higher International Normalised Ratio (INR), mortality end-stage liver disease (MELD) score, serum bilirubin and aspartate aminotransferase (AST) level. There was no significant difference in alanine aminotransferase (ALT), gamma-glutamyl transferase (GGT) and serum creatinine. Alcohol use disorder identification test (AUDIT) score was significantly higher in drinking controls compared to the cases (12 vs. 5, respectively) (Table 1).

### 3.2. Comparison of Global Profiling of the Serum microRNA Expression in Heavy Alcohol Users

We detected 125 serum microRNAs expressed above the threshold in heavy alcohol users (n = 47). An unsupervised hierarchical clustering heat map of expressed microRNAs differentiated cases from drinking controls (Figure 1). Twenty one microRNAs were significantly downregulated in cases compared to drinking controls (Figure 2a). These 21 microRNAs clearly differentiated cases from drinking controls, as visualised in the semi-supervised hierarchical clustering in heavy alcohol users Figure 2b.

A post hoc analysis of cases and drinking controls by sex, performed to test any sex-specific differences in miRNAs’ expression, showed 24 microRNAs were specific to females and 5 microRNAs were specific to males. (Appendix A). There were four common downregulated microRNAs (miR-27b, miR-221, miR-191, and miR-335) in both male and female cases compared to drinking controls.

### 3.3. microRNA Signature Distinguished Patients with Alcoholic Hepatitis

We then reallocated our cases to the alcoholic hepatitis (AH) group using the NIAAA criteria of a bilirubin cut-off of >3.0 mg/dL [18]. Cases with bilirubin < 3 mg/dL were considered as without hepatitis and defined as ‘cirrhosis’. In this post hoc analysis, to compare microRNA expression specific to AH, we compared AH (n = 12) to cirrhosis (n = 12), AH to drinking control and cirrhosis to drinking control (n = 23). Twenty-one miRNAs were downregulated in AH vs. drinking control (Figure 3a) and ten downregulated in cirrhosis vs. drinking control (Figure 3b). Further direct comparison of AH vs. cirrhosis cases identified three downregulated (miR-15a, miR-125b-5p and miR-185) and two upregulated (miR-21-5p and miR-28) microRNAs in AH group (Figure 3c).

Our main findings and post hoc results on microRNAs from different comparison groups are detailed in Appendix A. Drinking controls are defined as drinkers with no liver disease; alcoholic hepatitis (AH) is defined as cases with >3 mg/dL bilirubin); cirrhosis is defined as cases with <3 mg/dL bilirubin); cases are defined as drinkers with cirrhosis and AH and.

### 3.4. Correlation of Downregulated microRNAs with Traditional Biomarkers of Liver Disease

Seventeen microRNAs showed strong correlation with the clinical biomarkers (ALT, AST, Bilirubin, INR, MELD score, albumin, haemoglobin, platelet count and BMI). Highly significant positive correlation was seen with bilirubin, INR and MELD score, and inverse correlation with albumin and platelets count (Table 2 and Appendix A). The seven microRNAs (miR-16, miR-19a, miR-27a, miR-29b, miR-101, miR-130a and miR-191) had highly significant correlations with clinical biomarkers: bilirubin, INR and MELD score (*p* < 0.001, Table 2).

### 3.5. Diagnostic Performance of Conventional Biomarkers and Selected microRNAs to Distinguish Cases from Drinking Controls

Conventional markers INR, MELD, bilirubin and platelet count performed better (AUC-ROC 0.97, 0.95, 0.87 and 0.85, respectively) than miRNAs (Table 3). Of the seven microRNAs above, three microRNAs (miR-27a, miR-130a and miR-191) significantly predicted cases with AUC-ROC 0.8, 0.78 and 0.85, respectively (*p* < 0.020) (Table 3). The other microRNAs also predicted cases with moderate performance.

The three microRNAs also showed an inverse association trend of decreasing abundance with increasing disease severity (no disease, i.e., drinking control, alcoholic hepatitis and cirrhosis) but only miR-130a (*p* = 0.0412) and miR-191 (*p* = 0.0028) reached significance (Jonckheere–Terpstra nonparametric ordered test, Figure 4).

### 3.6. Association of miRNAs with Genetic Variants Linked to Risk of Cirrhosis

We performed correlation analysis of differentially expressed microRNAs in cases vs. drinking controls and risk allele dosages from available genotyping data from our previously identified variants associated with alcohol-associated cirrhosis (*PNPLA3*:rs2294915, *HSD17B13*:rs10433937, *FAF2*:rs11134977, *TM6SF2*:rs10401969, *SERPINA1*:rs28929474, *MBOAT7*:rs641738 and *MARC1*:rs2642438) [5]. Several microRNAs (miR-19a, miR-26a, miR-101, miR-151-3p, miR-221 and miR-301) showed a significant positive correlation with the *HSD17B13* variant (Appendix A). miR-27b, miR-130a and miR-335 had a significant positive correlation with single-nucleotide polymorphisms in *FAF2*:rs11134977, *TM6SF2*:rs10401969 and *MARC1*:rs2642438, respectively.

### 3.7. Identification of Target Genes and Pathway Analysis

We identified 1212 potential target messenger RNAs (mRNAs) for the 21 differentially expressed microRNAs specific to liver disease/functions using IPA. Co-analysis using significantly dysregulated microRNAs identified pathways for liver pathology (Figure 5). The liver-related pathways were enriched for liver inflammation/hepatitis, liver cirrhosis, liver fibrosis and liver steatosis (Figure 5). The most common ‘tox’ analysis pathways targeted by these microRNAs include fatty acid metabolism, liver necrosis/cell death, hepatic fibrosis, hepatic stellate cell activation, mitochondrial dysfunction, liver proliferation, increased liver damage, liver steatosis and liver hepatitis (Figure 6a). Several target mRNAs in disease/function pathways that regulate cirrhosis of liver, chronic hepatitis, hepatic steatosis, fibrosis of liver, synthesis of fatty acids, fatty acid metabolism and steatohepatitis were identified (Figure 6b). We also plotted the network diagram with seven microRNAs and the target mRNAs associated these microRNAs. One of the microRNA interaction networks had several mRNAs related to carbohydrate and lipid metabolism; for example, carbohydrate response element binding protein (*CREBP*), patatin-like phospholipase domain-containing protein 3 (*PNPLA3*) and both adipose triglyceride lipase (*ATGL*) and its activator α/β hydrolase domain containing 5 (*ABHD5*) (Figure 6a,b, Table 4). Similar pathway analysis for target mRNAs associated with AH-specific microRNAs revealed mRNAs related to inflammation, fibrosis, signalling and alcohol metabolism (Appendix A).

## 4. Discussion

One of the important findings of our study is the identification of a signature panel of microRNAs that specifically distinguished alcohol-associated liver cirrhosis from heavy alcohol users without liver disease. Unlike previous reports that compared ALD with healthy controls, our comparison avoided the chance of identifying miRNAs in response to alcohol exposure alone. Second, the abundance of a subset of microRNAs inversely correlated with progressive diseases stages (no disease, AH and cirrhosis) demonstrates their potential as indicators of disease severity. Third, among the downregulated microRNAs, seven (miR-16, miR-19a, miR-27a, miR-29b, miR-101, miR-130a and miR-191) exhibited a highly significant correlation with liver disease biomarkers, indicating their potential as robust biomarkers for liver cirrhosis in heavy drinkers. Interestingly, three microRNAs (miR-27a, miR-130a and miR-191) showed promising predictive accuracy in distinguishing cirrhosis cases from drinking controls, suggesting their capacity for clinical utility as diagnostic biomarkers for alcohol-associated liver cirrhosis. However, given that the conventional biomarkers were initially used to differentiate the cases from drinking controls, these traditional biomarkers with greater AUC-ROCs outperformed miRNAs, as expected.

As liver biopsy, the current gold standard for diagnosis, is invasive and carries certain limitations [19] less-invasive diagnostic biomarkers are preferred for identifying and monitoring liver cirrhosis [20]. Circulating microRNAs have emerged as promising biomarkers in various liver diseases, including hepatocellular carcinoma (HCC) and non-alcoholic fatty liver disease and steatohepatitis (NAFLD/NASH) [21,22,23,24]. Our protocol offers a convenient diagnostic approach using minimal volume of serum without the need for additional sample collection beyond the clinical requirement for treatment. The identification of our microRNAs distinguishing heavy drinkers with cirrhosis and their inverse association with disease severity offers a potential approach for diagnosing and monitoring the progression of this condition. Measuring the specific panel of downregulated microRNAs in serum samples may be developed as a reliable semi-quantitative and accessible means for detection of disease stages.

MicroRNAs have been previously implicated in various pathological and cellular pathways related to liver diseases [25,26,27,28]. Our pathway analysis revealed that the dysregulated microRNAs were associated with critical pathways involved in cell death (*Caspases*, *FASN*), fibrosis (*TGF*, *Col1*, *SMADs* and *MMPs*) inflammation (*IL6*, *TNF* and *PPARG*), steatosis and fatty acid metabolism (*ABHD* family, *ACOX1*, *ACOX3*, *HMGCR* and *LCLAT1*), specifically regulating triglyceride metabolism in the hepatic lipid droplets (*PNPLA2* (aka *ATGL*), *ABHD5* and *PNPLA3*).

Interestingly, the majority of SNPs identified as risk factors for alcohol-associated cirrhosis in recent GWAS reside in/around the genes related to lipid metabolism and lipid droplet biology (*PNPLA3*, *HSD17B13*, *FAF2* and *TM6SF2*) [5]. So, it was intriguing to find several target mRNAs, identified through disease-function and tox pathway networks, associated with triglyceride metabolism, particularly mRNAs *PNPLA3* (miR-29b and miR-27a), *PNPLA2*/*ATGL* (miR-148a, miR-27a) and *ABHD5* (miR-26a, miR-27a, miR-19b and miR-27b). *PNPLA3* is the strongest and the most replicated genetic risk for both alcohol- and non-alcohol-associated liver cirrhosis as reported in genome-wide association studies [5,6,29]. PNPLA3 exerts its effect by binding to α/β hydrolase domain 5 (ABHD5) dissociating it from adipose triglyceride lipase (ATGL), inactivating the major triglyceride lipase and resulting in triglyceride accumulation. However, we did not find significant correlation between any microRNA and the specific *PNPLA3*:rs2294915 risk variant. But, miR130a was significantly correlated with our novel discovery *FAF2*:rs11134977, another lipid droplet protein that binds to ATGL and also inhibits triglyceride metabolism. Furthermore, six microRNAs (miR-19a, miR-26a, miR-101, miR-151-3p, miR-221 and miR-301) had significant positive correlation with the *HSD17B13*:rs10433937 risk variant, another lipid-droplet-associated protein. The association of dysregulated miRNAs with target mRNAs that are involved in steatosis (lipid pathways, *SREMP*, *ACOX3*, *HMGCR* and *LCLAT1*), particularly triglyceride metabolism (*PNPLA3*, *ATGL ABDH* and *FAF2*) supports the new understanding that lipids play a key role in this disease pathogenesis and miRNAs may regulate this pathway, contributing to the underlying molecular mechanisms of lipid genetics in alcohol-associated cirrhosis. Future studies are needed to test this. Other target mRNAs involved in inflammation (*TNF, IFN, IL6* and *CXCL3*) and fibrosis (*TGFs* and *SMAD*) were also identified.

Surprisingly, in this study we did not identify the previous most commonly reported miRNAs (miR21, miR122 and miR155) associated with alcohol-associated cirrhosis, summarised in the two most-recent comprehensive reviews [30,31]. This may be because earlier studies were performed either in in vivo cross-species models or compared ALD/ALC with healthy controls. Our main comparison was within drinkers, with cirrhosis vs. drinkers without liver disease, and a relatively small cohort of AH. With reference to the above reviews, we did identify some common miRNAs previously identified in ALD (miR19, miR21-5p, miR26 and miR199), NAFLD (miR27b and miR30c) and viral hepatitis (miR27a, miR125b and 130a), suggesting common regulation in liver diseases. Our findings identifying several dysregulated miRNAs in cases (miR21-5p, miR29b, miR101, miR199a, miR221, miR335 and miR652) also reported in HCC [31], begs the speculation of a precancerous profile on pathway to HCC, as cirrhosis predominantly underlies the development of hepatocellular carcinoma.

## 5. Strengths and Limitations

A major strength of this study is the use of clinically and genetically well-characterised patients from the GenomALC-1 cohort. In addition, choosing drinkers as controls to compare liver cirrhosis patients is designed to minimise the specific effect of alcohol-associated microRNAs. However, we cannot rule out the possibility of bias towards alcohol-associated microRNAs in current drinkers, especially in the drinking control group. Global microRNA profiling of all three groups on the same array platform provides greater strength to our study for direct comparisons between groups and reliability of results.

One of our study limitations is that the sample size in our study was relatively small, and the results should be validated in larger cohorts and diverse populations. Additionally, the functional roles of the identified microRNAs in the pathogenesis of cirrhosis need to be elucidated through further experimental studies. We also acknowledge that microRNA expression profiles may have been affected in patients taking medications for liver disease and/or alcohol use disorder. Despite these limitations, our study has revealed interesting information on the involvement of microRNAs and their target mRNAs in identifying several pathways relevant in liver diseases. Most interestingly, our data identifying several lipid-related target genes support the recent genetic studies and school of thought regarding lipid biology as a key pathway in regulating liver diseases.

In conclusion, our findings highlight that microRNAs can have clinical relevance in their potential as diagnostic and/or therapeutic targets for alcohol-associated liver disease/cirrhosis. The study also underscores the role of triglyceride and lipid metabolism in this disease by identifying new miRNA and mRNA as potential therapeutic targets. Further research in this area may lead to the development of targeted therapeutic strategies for the management of liver diseases.

## Figures and Tables

**Figure 1 biology-12-01314-f001:**
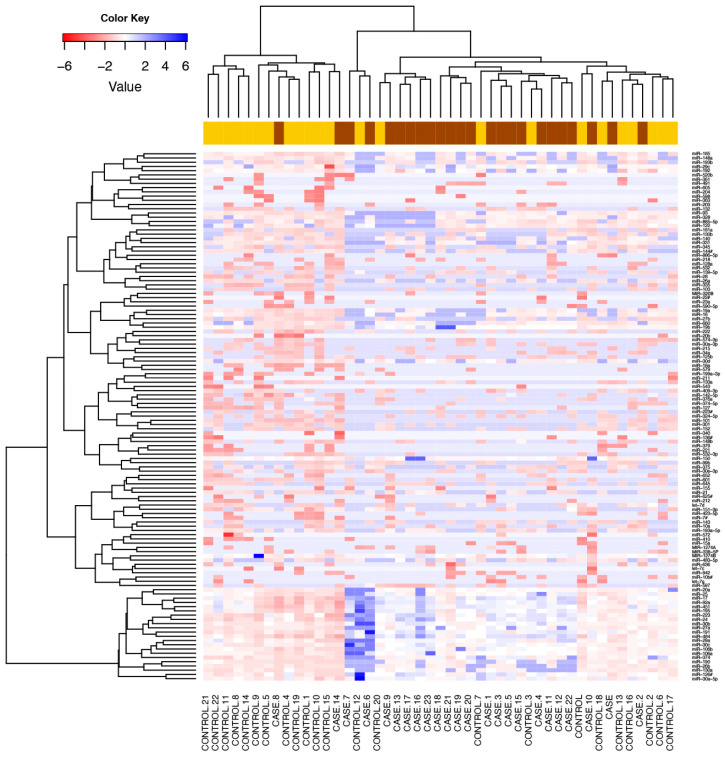
Differentially expressed microRNAs in cases versus drinking controls. Unsupervised hierarchical clustering heat map of expressed microRNAs differentiated cases (drinkers with liver cirrhosis) from drinking controls. Blue signals indicate lower abundance of microRNA in cases (brown) compared to drinking controls (yellow). Rows represent microRNAs and columns represent the individual samples. Colour key embedded in the figure shows the scale of microRNAs expression from blue (lower abundance) to red (higher abundance).

**Figure 2 biology-12-01314-f002:**
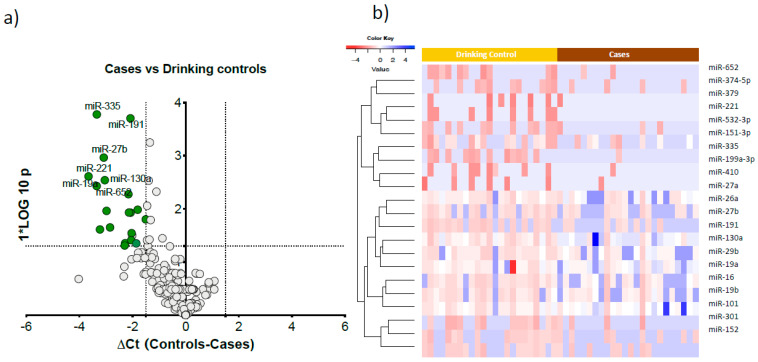
Significantly altered microRNAs in cases compared to drinking controls. (**a**). Volcano plot showing the significantly downregulated microRNAs in cases versus drinking controls. The *x*-axis is relative expression between controls and cases; the *y*-axis is *p* value plotted as −log10 scale. The green dots symbolise the significantly downregulated microRNAs (∆Ct > 1.5, *p* < 0.01) and grey dots represent non-significant microRNAs. (**b**). Semi-supervised hierarchical clustering heat map of 21 significantly expressed microRNAs differentiated cases from drinking controls. Colour key embedded in the figure shows the scale of microRNA expression from blue (lower abundance) to red (higher abundance).

**Figure 3 biology-12-01314-f003:**
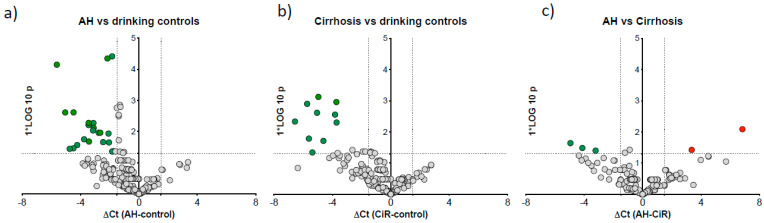
Differentially expressed microRNAs in AH compared to drinking controls and drinkers with cirrhosis. Volcano plots show differentially expressed microRNAs: (**a**) AH vs. drinking controls, (**b**) cirrhosis vs. drinking controls, and (**c**) AH vs. cirrhosis. The *x*-axis is relative expression (∆Ct); the *y*-axis is *p* value as −log_10_ scale. The red and green dots symbolise the significantly up- and downregulated microRNAs, respectively, based on *p* < 0.05 and ∆Ct value greater than 1.5; grey dots represent non-significant microRNAs.

**Figure 4 biology-12-01314-f004:**
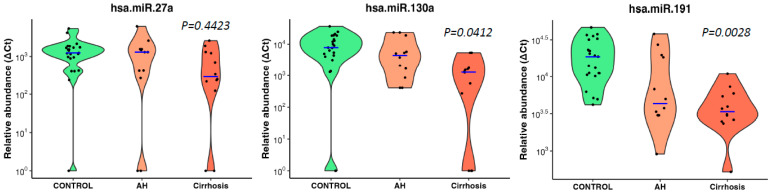
MicroRNAs associated with ALD severity in heavy drinkers. Violin plots show three significantly downregulated microRNAs showing high statistical correlation between decreasing abundance with increasing ALD severity in heavy drinkers. Blue lines indicate the medians, and each black dot represents a different sample in the cohort. The *y*-axis is relative abundance from the detectable limit (2^(39-Ct)). Ct of 39 is used as the threshold of detectability. microRNA expressions were normalised using two spike control ath-microRNAs (ath-miR-159a and ath-miR-172a). *p* is alternative *p* values from Jonckheere–Terpstra nonparametric ordered test. CONTROL—drinking controls, AH—drinkers with alcoholic hepatitis, Cirrhosis—drinkers with cirrhosis.

**Figure 5 biology-12-01314-f005:**
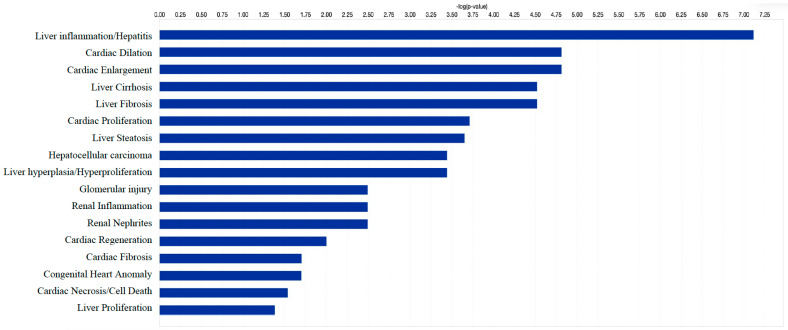
Significantly enriched pathways that are associated with 21 differentially expressed microRNAs. *x*-axis shows the negative log *p* value with threshold −log 0.05). Differentially expressed microRNAs show association mainly with liver-, cardiac- and renal-related disease/functional pathways. Co-analysis was performed in Ingenuity Pathway Analysis (IPA), (QIAGEN, Inc., https://targetexplorer.ingenuity.com).

**Figure 6 biology-12-01314-f006:**
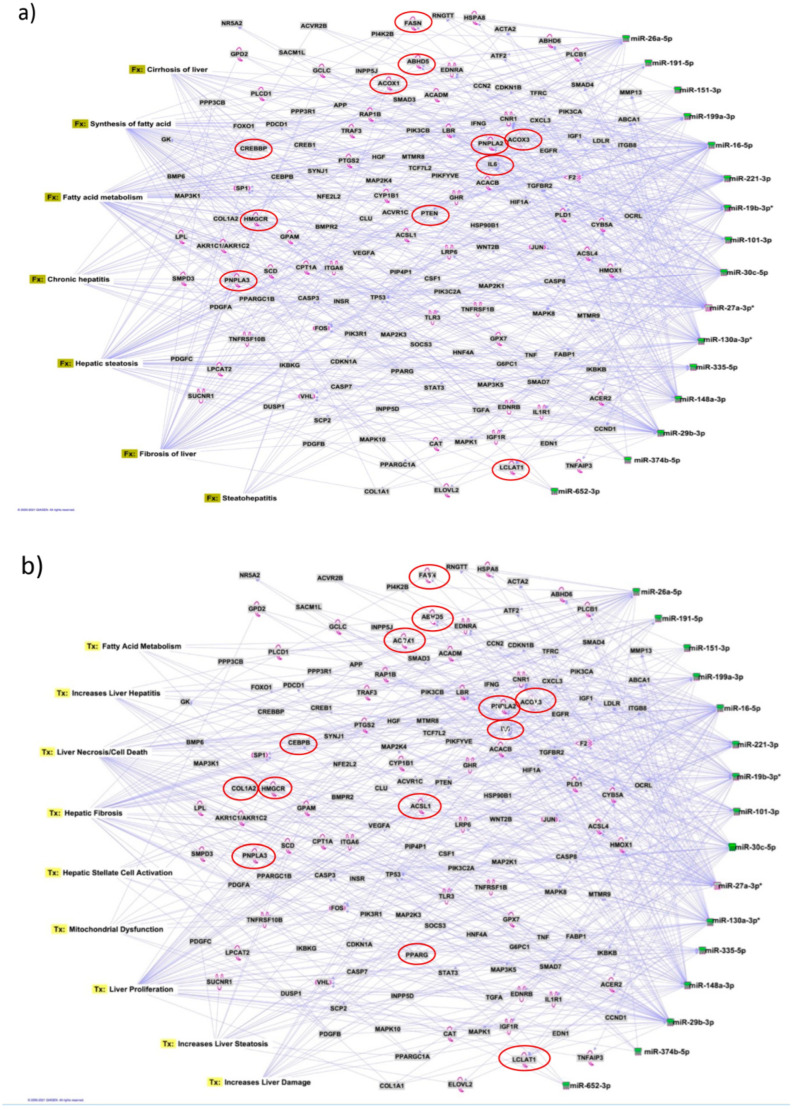
Pathway analysis identifies target mRNAs and microRNA-mRNA interactions. We used IPA to identify target mRNAs and microRNA-mRNA interactions. Target mRNAs were overlaid on (**a**) disease and functions (Fx) and (**b**) toxic functions (Tx) derived from select microRNAs using ‘Pathway design’ option in IPA. MicroRNA-targeted mRNA functions relevant to liver disease and lipid biology are circled in red. The top largest interactions of disease and toxic function pathway networks identified for microRNA–mRNA interactions are presented.

**Table 1 biology-12-01314-t001:** Clinical characteristics of participants.

		Cases (n = 24)		Drinking Controls (n = 23)		*p*-Value (Case Group vs. Drinking Control Group)
		Male n = 16	Female n = 8	Male n = 15	Female n = 8	
Demographics	European Caucasian ethnicity/race (%)	100%	100%	93%	100%	0.674
	Males (%)	66	34	65	35	0.864
	Age	51 (46–57)	49 (40–51)	56 (49–64)	52 (48–55)	0.156
Alcohol use	Alcohol intake (g/day)	236 ± 106	148 ± 89	228 ± 110	178 ± 84	0.779
	Years of high-risk drinking	30 (22–33)	21 (19–24)	22 (17–31)	16 (15–23)	0.133
	Life-time alcohol intake (Kg)	2497 ± 1542	1287 ± 1167	1952 ±1033	1267 ± 688	0.790
	Audit score	14 (2–33)	5 (3–21)	29 (29–36)	33 (29–33)	0.04 *
Lab results	Haemoglobin (g/L)	126 ± 20	118 ± 21	153 ±7.9	131 ± 6	0.001 *
	WBC	6.9 ± 2.7	6.0 ± 1.7	6.8 ± 2.7	8 ± 3	0.456
	Platelet count (10^9^/L)	151 ± 97	104 ± 37	254 ± 100	255 ± 72	<0.0001 *
	Albumin (g/L)	34 ± 12	35 ± 5	44 ± 3	45 ± 4	<0.0001 *
	Bilirubin (umol/L)	44 ± 37	62 ± 34	9 ± 5	8 ± 4	<0.0001 *
	Creatinine (mg/dL)	71 (58–94)	64 ± 21	82 ± 11	63 ±13	0.594
	ALT (IU/L)	43 ± 22	30 ± 10	38 ± 32	23 ± 11	0.623
	AST (IU/L)	74 ± 37	63 ± 40	35 ± 23	28 ±14	0.001 *
	GGT (U/L)	142 ± 127	127 ± 183	148 ± 181	62 ± 87	0.179
Liver disease severity	INR	1.3 ± 0.3	1.6 ± 0.5	1.0 ± 0.1	1.0 ± 0.1	<0.0001 *
	MELD score	10.1 ± 5.8	12 ± 5.2	2.3 ± 2.1	−0.7 ± 4.1	<0.0001 *

Data expressed as median (lower quartile–upper quartile) or n (%). N/A—not applicable. SCr refers to serum creatinine. * depicts the significance between the cases and controls.

**Table 2 biology-12-01314-t002:** Correlation analysis of miRNA expression (∆Ct) with traditional biomarkers.

miRs	ALT (IU/L)	AST (IU/L)	Bilirubin (mg/dL)	INR	MELD Score
**miR-16**	**0.01 (−0.2 to 0.30)**	**0.18 (−0.12 to 0.45)**	**0.39 ** (0.11 to 0.62)**	**0.51 *** (0.26 to 0.71)**	**0.48 *** (0.21 to 0.68)**
**miR-19a**	**0.26 (−0.03 to 0.51)**	**0.39 ** (0.11 to 0.61)**	**0.44 ** (0.16 to 0.65)**	**0.50 *** (0.24 to 0.69)**	**0.53 *** (0.27 to 0.71)**
miR-19b	0.11 (0.18 to 0.39)	0.23 (−0.06 to 0.49)	0.32 * (0.13 to 0.63)	0.36 * (0.19 to 0.66)	0.47 *** (0.20 to 0.67)
miR-26a	0.21 (−0.09 to 0.47)	0.27 (−0.03 to 0.52)	0.30 * (0.12 to 0.62)	0.30 * (0.01 to 0.54)	0.40 ** (0.15 to 0.64)
**miR-27a**	**0.13 (−0.17 to 0.40)**	**0.18 (−0.11 to 0.45)**	**0.38 ** (0.04 to 0.57)**	**0.39 ** (0.01 to 0.53)**	**0.39 ** (0.11 to 0.62)**
miR-27b	0.16 (−0.14 to 0.43)	0.28 (−0.01 to 0.53)	0.38 * (0.09 to 0.60)	0.41 ** (0.13 to 0.63)	0.36 * (0.07 to 0.59)
**miR-29b**	**0.24 (0.05 to 0.50)**	**0.20 (−0.09 to 0.47)**	**0.39 ** (0.07 to 0.59)**	**0.49 *** (0.22 to 0.68)**	**0.53 *** (0.27 to 0.71)**
miR-30c	0.08 (−0.21 to 0.37)	0.27 (−0.03 to 0.52)	0.36 * (0.18 to 0.66)	0.40 ** (0.12 to 0.62)	0.42 ** (0.26 to 0.71)
**miR-101**	**0.22 (−0.08 to 0.48)**	**0.30 * (0.01 to 0.54)**	**0.39 ** (0.04 to 0.57)**	**0.44 ** (0.16 to 0.64)**	**0.52 *** (0.27 to 0.71)**
**miR-130a**	**0.27 (0.02 to 0.52)**	**0.34 * (0.052 to 0.58)**	**0.43 ** (0.14 to 0.64)**	**0.53 *** (0.27 to 0.71)**	**0.43 ** (0.15 to 0.64)**
miR-151-3p	0.45 ** (0.18 to 0.65)	0.43 ** (0.16 to 0.65)	0.41 ** (0.13 to 0.63)	0.25 (0.01 to 0.51)	0.28 (−0.01 to 0.53)
**miR-191**	**0.29 * (0.01 to 0.54)**	**0.44 ** (0.16 to 0.65)**	**0.54 **** (0.29 to 0.72)**	**0.59 **** (0.35 to 0.75)**	**0.60 **** (0.37 to 0.76)**
miR-199a-3p	0.29 * (0.01 to 0.54)	0.39 ** (0.11 to 0.61)	0.33 * (0.04 to 0.57)	0.30 * (0.02 to 0.54)	0.32 * (0.02 to 0.56)
miR-221	0.17 (−0.13 to 0.44)	0.32 * (0.03 to 0.56)	0.50 *** (0.24 to 0.69)	0.35 * (0.06 to 0.58)	0.45 ** (0.17 to 0.66)
miR-335	0.13 (−0.16 to 0.41)	0.25 (−0.05 to 0.50)	0.36 * (0.06 to 0.59)	0.40 ** (0.12 to 0.62)	0.35 * (0.06 to 0.58)
miR-374-5p	0.08 (−0.36 to 0.21)	0.11 (−0.19 to 0.39)	0.40 ** (0.11 to 0.62)	0.34 * (0.04 to 0.57)	0.34 * (0.05 to 0.58)
miR-532-3p	0.15 (−0.15 to 0.42)	0.26 (−0.04 to 0.51)	0.34 * (0.05 to 0.58)	0.34 * (0.04 to 0.57)	0.39 ** (0.10 to 0.62)
miR-652	0.26 (−0.03 to 0.51)	0.42 ** (0.14 to 0.63)	0.33 * (0.04 to 0.57)	0.39 ** (0.11 to 0.61)	0.31 * (0.01 to 0.55)

Spearman correlation coefficients (r with IQR) are depicted with *p* values (* <0.05, ** <0.001, *** <0.0001, **** <0.00001). Seven microRNAs with correlation > 0.40 and *p* < 0.001 for bilirubin, INR and MELD biomarkers are bolded. ALT: alanine aminotransferase, AST: aspartate aminotransferase, INR: International Normalised Ratio, MELD: mortality end-stage liver disease.

**Table 3 biology-12-01314-t003:** Diagnostic performance of conventional biomarkers and selected microRNAs in distinguishing cases from controls.

		AUC-ROC (95%CI)	*p*-Value
Conventional biomarkers	INR	0.97 (0.92–1)	<0.0001
	MELD score	0.94 (0.87–1)	<0.0001
	Bilirubin (mg/dL)	0.89 (0.79–0.98)	<0.0001
	Platelet count	0.85 (0.74–0.96)	<0.0001
	Albumin/µmol/L	0.83 (0.70–0.96)	0.0001
	AST (U/L)	0.76 (0.62–0.90)	0.0023
MicroRNAs	**miR-191**	**0.85 (0.73–0.96)**	**<0.0001**
	**miR-27a**	**0.80 (0.67–0.93)**	**0.0004**
	**miR-130a**	**0.78 (0.64–0.92)**	**0.0009**
	miR-19a	0.77 (0.63–0.92)	0.0013
	miR-19b	0.76 (0.61–0.91)	0.0022
	miR-16	0.74 (0.59–0.89)	0.0050
	miR-29b	0.71 (0.56–0.87)	0.0128
	miR-101	0.68 (0.54–0.82)	0.0316

Diagnostic performance of AUC-ROCs greater than 0.65 are presented in the table (cases, n = 24 vs. controls, n = 23). MicroRNAs with AUC-ROC > 0.65 and *p* < 0.0001 are bolded. AST: aspartate aminotransferase, INR: International Normalised Ratio, MELD: mortality end-stage liver disease.

**Table 4 biology-12-01314-t004:** MicroRNA-targeted mRNA functions relevant to liver disease.

Target mRNA	Functions of Genes/Coded Molecule	miRNA Association from Pathway Analysis	Gene Name
*ABHD5*	Activates ATGL; dynamic interactions of ABHD5 with PNPLA3 regulate triacylglycerol metabolism in brown adipocytes	miR-26a, miR-27a, miR-19b, miR-27b	a/b hydrolase domain containing 5
*ABHD6*	Inactivation of ABHD6 protects against HFD-induced obesity, liver steatosis and insulin resistance	miR-151, miR-30c, miR-27a	a/b hydrolase domain containing 6
*ABHD3*	A lipase that selectively cleaves medium-chain and oxidatively truncated phospholipids	miR-130a, miR-221	a/b hydrolase domain containing 3
*ACOX1*	A rate-limiting enzyme in peroxisomal fatty acid β-oxidation, regulates metabolism, spontaneous hepatic steatosis and hepatocellular damage over time.	miR-199a, miR-16	Acyl-CoA Oxidase 1
*ACOX3*	Related to lipid metabolism	miR-151, miR-19a	Acyl-CoA Oxidase 3
*CREBP*	Key modulator of glycolytic, lipogenic and microsomal triglyceride transfer protein (Mttp) gene expression, thereby controlling both fatty acid accumulation and VLDL export from the liver	miR-191, miR-26a	carbohydrate response element binding protein
*HMGCR*	Transmembrane glycoprotein that is the rate-limiting enzyme in cholesterol biosynthesis as well as in the biosynthesis of nonsterol isoprenoids that are essential for normal cell function, including ubiquinone and geranylgeranyl proteins	miR-29b, miR-27a	3-hydroxy-3-methyl-glutaryl-coenzyme A reductase
*IL6*	Liver inflammation	miR-26a, miR-148a, miR-19b, miR-16, miR-30c, miR-191	Interleukin 6
*LBR*	Cholesterol synthesis via sterol reductase function	miR-221, miR-101, miR-27a, miR-130a	Lamin B receptor
*PNPLA2/ATGL*	The key enzyme for intracellular hydrolysis of stored triglycerides and determines FA signalling through PPARα; we explored the role of ATGL in hepatic inflammation in mouse models of NASH and endotoxemia	miR-148a, miR-27a	Adipose triglyceride lipase
*PNPLA3*	Exhibits a dual function in LD metabolism, and that it participates in the restoration of lipid	miR-29b, miR-27a	Patatin-like phospholipase domain-containing protein 3
*TCF7L2*	Susceptibility gene for type-2 diabetes	miR-191, miR-26a, miR-221	Transcription factor

## Data Availability

The data presented in this study are available on request from the corresponding author. The data are not publicly available due to privacy issues.

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
