# Peer review of "MicroRNAs Signature Panel Identifies Heavy Drinkers with Alcohol-Associated Cirrhosis from Heavy Drinkers without Liver Injury"

_biology, 2023, doi:10.3390/biology12101314_

Round 1

Reviewer 1 Report

The authors try to identify microRNAs involved in the development of cirrhosis of alcoholic origin, to establish future therapeutic measures and understand the evolution of the disease.

1. All abbreviations should be included in the manuscript the first time they are cited. M? F?

2. How the healthy controls (n=5) differ from the controls (n=23)

3. Have the authors verified that the miRNAs detected are also present in patients who develop cirrhosis without alcohol consumption or viral infection?

4. The sample size should be increased, I consider that 5 healthy controls are not enough to establish a conclusion and make comparisons.

5. The authors indicate a 10% progression to hepatocellular carcinoma in the introduction. This figure is very high, the authors should indicate if it is cirrhosis due to HCV or due to alcohol.

6. The introduction could be expanded by introducing recent studies on genetic markers involved in alcoholic cirrhosis.

7. The authors talk about the need for non-invasive diagnosis, however they contemplate the use of the widely used fibroscan.

8. The normalized values of all biochemical parameters should be included according to the population analyzed, making a distinction between men and women.

9. All table abbreviations should be included in the table footer.

10. All variables should be expressed in terms of mean and SD.

11. The amount of cirrhotic alcohol varies according to sex. The authors mix both sexes although there is a larger population of men. In table 1 you should see 17% in the healthy control (n=5).

12. Table 1. P value, what comparisons are made? There are three groups…

This aspect should be indicated. The healthy control values should be included, as well as the reference values in the population.

13. section 3.2. Do the authors refer to heavy alcohol users? Do these patients have cirrhosis? The terminology to define the 3 groups analyzed should be homogeneous throughout the text to avoid misunderstandings.

14. section 3.2. heavy alcohol users n=47, does not coincide with what is indicated in table 1.

15. The authors indicate 77 SDRs, however they add up to 82. Could you clarify this aspect?

16. Fig 1. Alcohol user…with or without cirrhosis??? They should change the title which leads to misunderstanding. They must also indicate the rest of the colors of the scale in the figure caption.

17. Fig 3rd, it is impossible to visualize the 21 microRNAs.

18. MELD score values should be indicated in the methodology and referenced.

19. Table 2. It is not well understood, because there are biomarkers marked in bold and others not despite being statistically significant.

20. Table 2. Platelets are NOT analyzed, nor are the rest of the variables that appear in Table 1. Could you indicate the reason for this selection?

21. Fig4, best? Or highly statistically correlated?

22. fig 7. Fx?

23. Patients with cirrhosis should be indicated whether or not they are taking any medication that could interfere with the analysis, and the degree of fibrosis (F1-F4) should also be indicated.

Reviewer 2 Report

 1.       Regarding your controls, please clearly state if you considered the patients with ALD (but no cirrhosis) or heavy drinkers with no liver changes as controls. In this form is confusing. Please maintain the concept uniformity throughout the manuscript

2.       Please define the syntagm no significant liver disease.

3.       I would recommend expressing the comparative results between patients with HC and without, otherwise might be confusing (i.e. abstract-results section- alcohol users compared with HC). The subgroups should be uniformly named throughout the manuscript: control, heavy drinkers without HC and heavy drinkers with HC.

4.       In the introduction you mentioned ALD. Did you include here also HC? The aim of the study meant in the introduction is not in accordance with the summary and abstract. Please decide which are the main and secondary objectives of your study.

5.       Methods: Please define the group including heavy drinkers without significant liver disease (i.e., fibrosis? etc.)

6.       Results. Please redefine the 3 groups used and mention them uniformly through the manuscript (i.e., controls, heavy drinkers without HC and HC)

7.       Results 3.2. Please consider to mention the differences between patients with HC and patients with HC and AH. How did you diagnosticate the AH in patients with HC? Reconsider the recommendation no.6

8.       Fig.2. Please reconsider the groups in accordance with the group’s definition (i.e. fig.2c, HC vs cases- does cases not include patients with HC?)

9.       Fig.5. It is even more confusing the new group: acute hepatitis?!

10.   The Discussion section should be focused on the main groups considered (redefined) and their results respectively 

1

Round 2

Reviewer 1 Report

The authors have improved the manuscript and have responded adequately to the questions raised, so I recommend acceptance of the manuscript in its current form.